# Solving Continuous Control via Q-learning

**Tim Seyde**[*]
MIT CSAIL

**Peter Werner**
MIT CSAIL

**Wilko Schwarting**
MIT CSAIL

**Igor Gilitschenski**
University of Toronto

**Martin Riedmiller**
DeepMind

**Daniela Rus**[1]
MIT CSAIL

**Markus Wulfmeier**[1]
DeepMind

## Abstract

While there has been substantial success for solving continuous control with actor-critic methods, simpler critic-only methods such as Q-learning find limited application in the associated high-dimensional action spaces. However, most actor-critic methods come at the cost of added complexity: heuristics for stabilisation, compute requirements and wider hyperparameter search spaces. We show that a simple modification of deep Q-learning largely alleviates these issues. By combining bang-bang action discretization with value decomposition, framing single-agent control as cooperative multi-agent reinforcement learning (MARL), this simple critic-only approach matches performance of state-of-the-art continuous actor-critic methods when learning from features or pixels. We extend classical bandit examples from cooperative MARL to provide intuition for how decoupled critics leverage state information to coordinate joint optimization, and demonstrate surprisingly strong performance across a variety of continuous control tasks. [2]

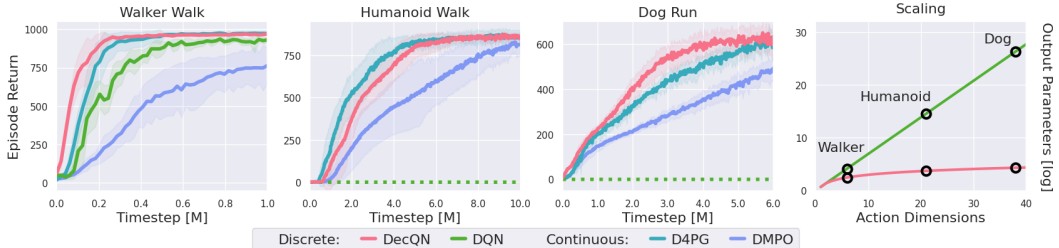

Figure 1: Q-learning yields state-of-the-art performance on various continuous control benchmarks. Simply combining bang-bang action discretization with full value decomposition scales to high-dimensional control tasks and recovers performance competitive with recent actor-critic methods. Our Decoupled Q-Networks (DecQN) thereby constitute a concise baseline agent to highlight the power of simplicity and to help put recent advances in learning continuous control into perspective.

## 1 Introduction

Reinforcement learning provides a powerful framework for autonomous systems to acquire complex behaviors through interaction. Learning efficiency remains a central aspect of algorithm design, with a broad spectrum spanning sample-efficient model-based off-policy approaches (Ha & Schmidhuber, 2018; Hafner et al., 2019) at one extreme and time-efficient on-policy approaches leveraging parallel simulation at the other extreme (Rudin et al., 2022; Xu et al., 2021). Particularly in high-dimensional domains with complicated environment dynamics and task objectives, complex trade-offs between representational capacity, exploration capabilities, and optimization accuracy commonly arise.

Continuous state and action spaces yield particularly challenging exploration problems due to the vast set of potential trajectories they induce. Significant research effort has focused on improving

---

[*]Correspondence to `tseyde@mit.edu`. [1]Equal advising.
[2]Project website: `https://sites.google.com/view/decoupled-q-networks/home`

efficiency through representation learning in the context of model-free abstraction or model-based planning (Ha & Schmidhuber, 2018; Srinivas et al., 2020; Wulfmeier et al., 2021), guided exploration via auxiliary rewards (Osband et al., 2016; Pathak et al., 2017; Sekar et al., 2020; Seyde et al., 2022b), or constrained optimization particularly to stabilize learning with actor-critic approaches (Schulman et al., 2015; Haarnoja et al., 2018; Abdolmaleki et al., 2018). However, recent results have shown that competitive performance can be achieved with strongly reduced, discretized versions of the original action space (Tavakoli et al., 2018; Tang & Agrawal, 2020; Seyde et al., 2021).

This opens the question whether tasks with complex high-dimensional action spaces can be solved using simpler critic-only, discrete action-space algorithms instead. A potential candidate is Q-learning which only requires learning a critic with the policy commonly following via $\epsilon$-greedy or Boltzmann exploration (Watkins & Dayan, 1992; Mnih et al., 2013). While naive Q-learning struggles in high-dimensional action spaces due to exponential scaling of possible action combinations, the multi-agent RL literature has shown that factored value function representations in combination with centralized training can alleviate some of these challenges (Sunehag et al., 2017; Rashid et al., 2018), further inspiring transfer to single-agent control settings (Sharma et al., 2017; Tavakoli, 2021). Other methods have been shown to enable application of critic-only agents to continuous action spaces but require additional, costly, sampling-based optimization (Kalashnikov et al., 2018).

We build on insights at the intersection of these methods to show that a surprisingly straightforward variation of deep Q-learning (Mnih et al., 2013), within the framework of Hypergraph Q-Networks (HGQN) (Tavakoli et al., 2021), can indeed solve various state- and pixel-based single-agent continuous control problems at performance levels competitive with state-of-the-art continuous control algorithms. This is achieved by a combination of extreme action space discretization and full value decomposition with extensive parameter-sharing, requiring only small modifications of DQN.

To summarize, this work focuses on the following key contributions:

- The DecQN agent as a simple, decoupled version of DQN combining value decomposition with bang-bang action space discretization to achieve performance competitive with state-of-the-art continuous control actor-critic algorithms on state- and pixel-based benchmarks.

- The related discussion of which aspects are truly required for competitive performance in continuous control as bang-bang control paired with actuator decoupling in the critic and without an explicit policy representation appears sufficient to solve common benchmarks.

- An investigation of time-extended collaborative multi-agent bandit settings to determine how decoupled critics leverage implicit communication and the observed state distribution to resolve optimisation challenges resulting from correlations between action dimensions.

## 2 RELATED WORKS

**Discretized Control** Continuous control problems are commonly solved with continuous policies (Schulman et al., 2017; Abdolmaleki et al., 2018; Haarnoja et al., 2018; Hafner et al., 2019; Yarats et al., 2021). Recently, it has been shown that even discretized policies can yield competitive performance and favorable exploration in continuous domains with acceleration-level control (Tavakoli et al., 2018; Farquhar et al., 2020; Neunert et al., 2020; Tang & Agrawal, 2020; Seyde et al., 2021; 2022a). When considering discretized policies, discrete action-space algorithms are a natural choice (Metz et al., 2017; Sharma et al., 2017; Tavakoli, 2021). Particularly Q-learning based approaches promise reduced model complexity by avoiding explicit policy representations (Watkins & Dayan, 1992), although implicit policies in the form of proposal distributions may be required for scalability (Van de Wiele et al., 2020). We build on perspectives from cooperative multi-agent RL to tackle complex single-agent continuous control tasks with a decoupled extension of Deep Q-Networks (Mnih et al., 2013) over discretized action spaces to reduce agent complexity and to better dissect which components are required for competitive agents in continuous control applications.

**Cooperative MARL** Conventional Q-learning requires both representation and maximisation over an action space which exponentially grows with the number of dimensions and does not scale well to the high-dimensional problems commonly encountered in continuous control of robotic systems. Significant research in multi-agent reinforcement learning (MARL) has focused on improving scalability of Q-learning based approaches (Watkins & Dayan, 1992). Early works considered indepen-

dent learners (Tan, 1993), which can face coordination challenges particularly when operating under partial observability (Claus & Boutilier, 1998; Matignon et al., 2012). Some methods approach these challenges through different variations of optimistic updates (Lauer & Riedmiller, 2000; Matignon et al., 2007; Panait et al., 2006), while importance sampling can assist with off-environment learning (Foerster et al., 2017). Furthermore, different degrees of centralization during both training and execution can address non-stationarity with multiple actors in the environment (Busoniu et al., 2006; Lowe et al., 2017; Böhmer et al., 2019). We take a cooperative multi-agent perspective, where individual actuators within a single robot aim to jointly optimize a given objective.

**Value Decomposition**  One technique to avoid exponential coupling is value function factorization. For example, interactions may be limited to subsets of the state-action space represented by a coordination graph (Guestrin et al., 2002; Kok & Vlassis, 2006). Schneider et al. (1999) consider a distributed optimization by sharing return information locally, while Russell & Zimdars (2003) share value estimates of independent learners with a centralized critic. Similarly, Yang et al. (2018) factorize value functions across interactions with neighbors to determine best response behaviors. Sunehag et al. (2017) introduced Value-Decomposition Networks (VDNs) to approximate global value functions as linear combinations of local utility functions under partial observability. Rashid et al. (2018) extend this to non-linear combinations in QMIX, where a centralized critic further leverages global state information. Son et al. (2019) propose QTRAN to factorizes a transformation of the original value function based on a consistency constraint over a joint latent representation. These insights can further be transferred to the multi-agent actor-critic setting (Wang et al., 2020; Su et al., 2021; Peng et al., 2021). Our approach scales to complex morphologies by leveraging a linear factorization over action dimensions within a centralized critic conditioned on the global robot state.

**Parameter Sharing**  Efficiency of learning a factored critic can improve through parameter-sharing (Gupta et al., 2017; Böhmer et al., 2020; Christianos et al., 2021) and non-uniform prioritization of updates (Bargiacchi et al., 2021). In particular, the Hybrid Reward Architecture (HRA) proposed by Van Seijen et al. (2017) provides a compact representation by sharing a network torso between individual critics (see also (Chu & Ye, 2017)). We achieve efficient learning at reduced model complexity through extensive parameter-sharing within a joint critic that splits into local utility functions only at the output neurons, while biasing sampling towards informative transitions through prioritized experience replay (PER) on the temporal-difference error (Schaul et al., 2015).

**Decoupled Policies**  Stochastic policies for continuous control are commonly represented by Gaussians with diagonal covariance matrix, e.g. (Schulman et al., 2017; Haarnoja et al., 2018; Abdol-maleki et al., 2018), while we leverage decoupled Categoricals as a discrete counterpart. Fully decoupled discrete control via Q-learning was originally proposed by Sharma et al. (2017) to separate dimensions in Atari. Tavakoli et al. (2021) extend this concept to hypergraph Q-networks (HGQN) to leverage mixing across higher-order action subspaces (see also Appendix F). Here, we demonstrate that learning bang-bang controllers, leveraging only extremal actions at the action space boundaries and inclusion of the zero action for bang-off-bang control (Bellman et al., 1956; LaSalle, 1960), with full decoupling as a minimal modification to the original DQN agent yields state-of-the-art performance across feature- and pixel-based continuous control benchmarks. Our investigation provides additional motivation for existing work and proposes a straightforward if surprising baseline to calibrate continuous control benchmark performance by pushing simple concepts to the extreme.

## 3  BACKGROUND

We describe the reinforcement learning problem as a Markov Decision Process (MDP) defined by the tuple $\{\mathcal{S}, \mathcal{A}, \mathcal{T}, \mathcal{R}, \gamma\}$, where $\mathcal{S} \subset \mathbb{R}^N$ and $\mathcal{A} \subset \mathbb{R}^M$ denote the state and action space, respectively, $\mathcal{T} : \mathcal{S} \times \mathcal{A} \to \mathcal{S}$ the transition distribution, $\mathcal{R} : \mathcal{S} \times \mathcal{A} \to \mathbb{R}$ the reward mapping, and $\gamma \in [0, 1)$ the discount factor. We define $s_t$ and $a_t$ to be the state and action at time $t$ and represent the policy by $\pi(a_t|s_t)$. The discounted infinite horizon return is given by $G_t = \sum_{\tau=t}^{\infty} \gamma^{\tau-t} R(s_\tau, a_\tau)$, where $s_{t+1} \sim \mathcal{T}(\cdot|s_t, a_t)$ and $a_t \sim \pi(\cdot|s_t)$. Our objective is to learn the optimal policy maximizing the expected infinite horizon return $\mathbb{E}[G_t]$ under unknown transition dynamics and reward mappings.

Figure 2: Decoupled Q-Network (DecQN) with bang-bang actions (2 bins per action dimension). The network predicts one state-action value for each decoupled action based on the observed state. During training, selection proceeds either via indexing for the observed actions at the current timestep (green) or via decoupled maximization along each action dimension at the next timestep. A simple $\epsilon$-greedy policy is recovered based on the decoupled argmax over the $Q$-function.

## 3.1 POLICY REPRESENTATION

Current state-of-the-art algorithms for continuous control applications commonly consider the actor-critic setting with the policy modelled as a continuous distribution $\pi_\phi(a_t|s_t)$ independently from the value estimator $Q_\theta(s_t, a_t)$, or $V_\theta(s_t)$, both represented by separate neural networks with parameters $\phi$ and $\theta$. Recent results have shown that comparable performance can be achieved with these approaches when replacing the continuous policy distributions by discretized versions (Seyde et al., 2021), at the potential cost of principled ways to represent stochastic policies. One may then ask whether we require the actor-critic setting or if simpler discrete control algorithms are sufficient. Q-learning presents itself as a lightweight alternative by only learning the value function and recovering the policy by $\epsilon$-greedy or Boltzmann exploration, side-stepping explicit policy gradients.

## 3.2 DEEP Q-NETWORKS

We consider the framework of Deep Q-Networks (DQN) (Mnih et al., 2013), where the state action value function $Q_\theta(s_t, a_t)$ is represented by a neural network with parameters $\theta$. The parameters are updated in accordance with the Bellman equation by minimizing the temporal-difference (TD) error. We leverage several modifications that accelerate learning and improve stability based on the Rainbow agent (Hessel et al., 2018) as implemented in Acme Hoffman et al. (2020).

In particular, we leverage target networks for improved stability in combination with double Q-learning to mitigate overestimation (Mnih et al., 2015; Van Hasselt et al., 2016). We further employ prioritized experience replay (PER) based on the temporal difference error to bias sampling towards more informative transitions and thereby accelerate learning (Schaul et al., 2015). Additionally, we combine rewards over several consecutive transitions into multi-step return to improve stability of Bellman backups (Sutton & Barto, 2018). A more detailed discussion of these aspects is included in Appendix G, while we provide ablation studies on individual components in Appendix E.

Combining these modification of the original DQN agent, we optimize the following loss function

$$\mathcal{L}(\theta) = \sum_{b=1}^{B} L_\delta(y_t - Q_\theta(s_t, a_t)), \tag{1}$$

where action evaluation employs the target $y_t = \sum_{j=0}^{n-1} \gamma^j r(s_{t+j}, a_{t+j}) + \gamma^n Q_{\theta-}\left(s_{t+n}, a_{t+n}^*\right)$, action selection uses $a_{t+1}^* = \arg\max_a Q_\theta(s_{t+1}, a)$, $L_\delta(\cdot)$ is the Huber loss and the batch size is $B$.

## 4 DECOUPLED Q-NETWORKS

The Rainbow agent (Hessel et al., 2018) from the previous section provides a framework for learning high-quality discrete control policies. However, conventional Q-learning methods enumerate the action space and can therefore become inefficient when attempting to scale to high dimensions.

### 4.1 MULTI-AGENT PERSPECTIVE

Scaling Q-learning to high-dimensional action spaces has been studied extensively in MARL settings. Particularly in cooperative MARL it is common to consider factorized value functions in combination with centralized training and decentralized execution (CTDE). Factorization reduces the effective dimensionality from the perspective of individual agents by learning utility functions conditioned on local observations. Centralized training considers global information by composing individual utilities and optimizing for consistent behavior among agents. Decentralized execution then assumes that the local utility functions are sufficient for optimal distributed action selection.

### 4.2 FACTORIZED $Q$-FUNCTION

Inspired by the MARL literature, we consider our agent to be a team of individual actuators that aim to cooperatively solve the given objective. We further assume that the overall state-action value function $Q(\boldsymbol{s}_t, \boldsymbol{a}_t)$ can be locally decomposed along action dimensions into a linear combination of $M$ single action utility functions $Q_\theta^j(\boldsymbol{s}_t, a_t^j)$. This simply applies the principle of value decomposition (Sunehag et al., 2017) and fits within the general concept of HGQN-type algorithms (Tavakoli et al., 2021) as a simplified variation of the base case without a hypergraph as was also leveraged by Sharma et al. (2017) for Atari. The overall state-action value function is then represented as

$$Q_\theta(\boldsymbol{s}_t, \boldsymbol{a}_t) = \sum_{j=1}^{M} \frac{Q_\theta^j(\boldsymbol{s}_t, a_t^j)}{M}, \tag{2}$$

where each utility function is conditioned on the global robot state to facilitate actuator coordination via implicit communication as we observe in our experiments. This is further assisted by a high-degree of parameter sharing within a unified critic (Van Seijen et al., 2017) that only splits to predict decoupled state-action utilities at the outputs. The linear combination of univariate utility functions then allows for efficient decomposition of the argmax operator

$$\arg\max_{\boldsymbol{a}_t} Q_\theta(\boldsymbol{s}_t, \boldsymbol{a}_t) = \left( \arg\max_{a_t^1} Q_\theta^1(\boldsymbol{s}_t, a_t^1), \dots, \arg\max_{a_t^M} Q_\theta^M(\boldsymbol{s}_t, a_t^M) \right), \tag{3}$$

where each actuator only needs to consider its own utility function. Global optimization over $\boldsymbol{a}_t$ then simplifies into parallel local optimizations over $a_t^j$. Training still considers the entire robot state and all joint actions within the centralized value function, while online action selection is decoupled. Inserting this decomposition into the Bellman equation yields a decoupled target representation

$$y_t = r(\boldsymbol{s}_t, \boldsymbol{a}_t) + \gamma \sum_{j=1}^{M} \max_{a_{t+1}^j} \frac{Q_\theta^j(\boldsymbol{s}_{t+1}, a_{t+1}^j)}{M}. \tag{4}$$

We can then insert the factorized value function of Eq. 2 and the decoupled target of Eq. 4 into Eq. 1. We also considered a composition based on learned weights inspired by QMIX (Rashid et al., 2018). Our findings suggest that access to the robot state throughout local utility functions outweighs the potential gains of more complex combinations, see also (Tavakoli et al., 2021) for a more detailed discussion on monotonic mixing functions. The following section further discusses the importance of shared observations and state-space coordination based on illustrative matrix games.

## 5 EXPERIMENTS

The DecQN agent solves various continuous control benchmark problems at levels competitive with state-of-the-art continuous control algorithms. First, we provide intuition for how DecQN reacts to coordination challenges based on illustrative matrix games. Then, we show results for learning state-based control on tasks from the DeepMind Control Suite (Tunyasuvunakool et al., 2020) and MetaWorld (Yu et al., 2020) - including tasks with action dimension $dim(\mathcal{A}) = 38$ - and compare to the state-of-the-art DMPO and D4PG agents (Abdolmaleki et al., 2018; Bellemare et al., 2017; Barth-Maron et al., 2018). We further provide results for pixel-based control on Control Suite tasks and compare to the state-of-the-art DrQ-v2 and Dreamer-v2 agents (Yarats et al., 2021; Hafner et al., 2020). Lastly, we discuss qualitative results for learning a velocity-tracking controller on a simulated Mini Cheetah quadruped in Isaac Gym (Makoviychuk et al., 2021) highlighting DecQN's versatility.

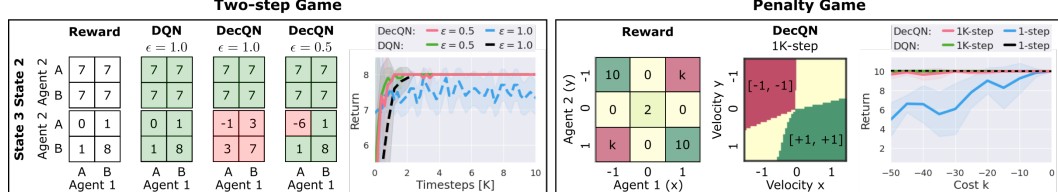

Figure 3: DecQN and DQN on cooperative matrix games. Left: a two-step game where agent 1 selects in step 1 which payoff matrix is used in step 2 (top vs. bottom). Learned Q-values of DecQN indicate that accurate values around the optimal policy are sufficient ($\epsilon = 0.5$) even when the full value distribution cannot be represented well ($\epsilon = 1.0$). Right: matrix game with actions as acceleration input to pointmass (x vs. y). DecQN struggles to solve the 1-step game (no dynamics). In the multi-step case, DecQN leverages velocity information to coordinate action selection (middle).

## 5.1 ILLUSTRATIVE EXAMPLES: COORDINATION IN MATRIX GAMES

Representing the critic as a linear decomposition across agents enables scaling to high-dimensional tasks while limiting the ability to encode the full value distribution (Sunehag et al., 2017; Rashid et al., 2018). This can induce coordination challenges (Lauer & Riedmiller, 2000; Matignon et al., 2012). Here, we consider cooperative matrix games proposed by Rashid et al. (2018) and Claus & Boutilier (1998) to illustrate how DecQN still achieves coordination in a variety of settings.

**Two-step game:** We look at a 2-player game with three states proposed by Rashid et al. (2018). In state 1, agent 1 playing action A/B transitions both agents to state 2/3 without generating reward. In states 2 or 3, the actions of both agents result in the cooperative payout described in Figure 3 (left). Rashid et al. (2018) showed that a VDN-type decomposition is unable to learn the correct value distribution under a uniform policy, which our results confirm for DecQN with $\epsilon = 1.0$ (middle). However, DecQN solves the task for non-uniform sampling with $\epsilon = 0.5$ by learning an accurate value distribution around the optimal policy (right). In online RL settings, agents actively shape the data distribution and learning to differentiate behavior quality is often sufficient for solving a task.

In the following we consider cooperative 2-player matrix games based on Claus & Boutilier (1998). We further augment the tasks by point-mass dynamics, where each agent selects acceleration input along the x or y-axis. We evaluate three variations: "1-step" resets after 1 step (no dynamics); "1K-step" resets after 1000 steps (dynamics); "Optimism" runs "1K-step" with optimistic updating inspired by Lauer & Riedmiller (2000). Each agent selects from three actions and we set $\epsilon = 0.5$.

**Penalty game:** The reward structure is defined in action-space and provided in Figure 3 (right). The two maxima at $(a_1, a_2) = \{(-1, -1), (+1, +1)\}$ and minima at $(a_1, a_2) = \{(-1, +1), (+1, -1)\}$ induce a Pareto-selection problem that requires coordination. DecQN is unable to consistently solve the 1-step game for high costs $k$ while it does solve the multi-step variation. In the latter, we find that DecQN coordinates actions (acceleration) based on state information (velocity). Computing action outputs over varying velocity inputs indicates a clear correlation in directionality across the unambiguous velocity pairings (middle). This underlines the importance of shared observations to achieve coordination with decoupled agents.

**Climbing game:** The reward structure is defined in Figure 4 with action (top) and state space (bottom) rewards. In action space, Nash equilibria at $(a_1, a_2) = \{(-1, -1), (0, 0)\}$ are shadowed by action $(a_1, a_2) = \{(1, 1)\}$, which has a higher minimum gain should one agent deviate (Matignon et al., 2012). DecQN often settles at the sub-optimal action $(a_1, a_2) = (0, 0)$ to avoid potential cost of unilateral deviation, which can be resolved by optimistic updating. However, this can impede learn-

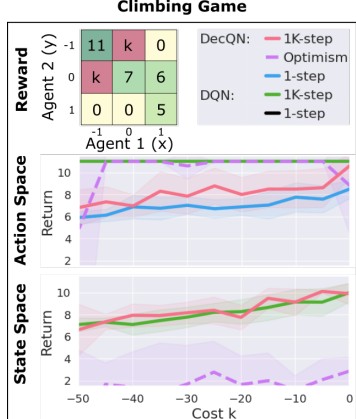

Figure 4: Climbing game with action (top) or state space (bottom) reward. Optimism helps DecQN in action space, while DecQN matches DQN performance under state space reward.

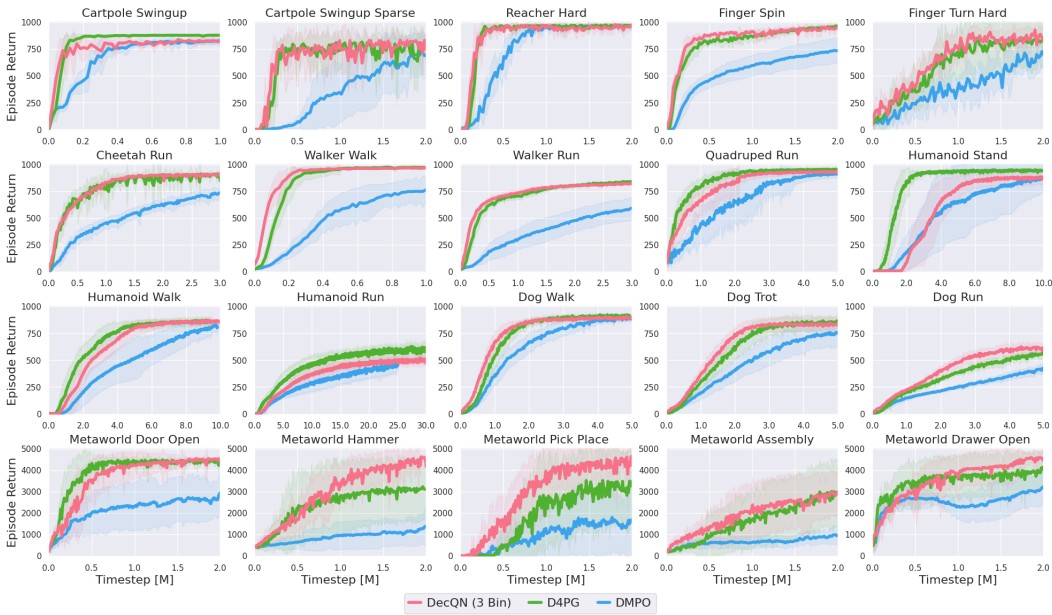

Figure 5: Performance for state-based control on DeepMind Control Suite and MetaWorld tasks. We compare DecQN with a discrete bang-off-bang policy to the continuous D4PG and DMPO agents. Mean and standard deviation are computed over 10 seeds with a single set of hyperparameters. DecQN yields performance competitive with state-of-the-art continuous control agents, scaling to high-dimensional Dog tasks via a simple decoupled critic representation and an $\epsilon$-greedy policy.

ing in domains that require state-space exploration as highlighted for state-space reward (bottom). We there find DecQN and DQN to achieve similar performance. In fact, the tasks considered in this paper do not require coordination at the action level but rather in higher-level integrated spaces.

Generally, our objective is learning control of articulated agents, which requires temporarily-extended coordination in state space. We find that while a decoupled critic may not be able to represent the full action-value distribution, it is often sufficient to differentiate action quality around the optimal policy. In particular, centralized learning combined with shared observations enables actuator coordination even in complex, high-dimensional domains as we observe in the next section.

## 5.2 BENCHMARKING: CONTINUOUS CONTROL FROM STATE INPUTS

We evaluate performance of the DecQN agent on several continuous control environments from the DeepMind Control Suite as well as manipulation tasks for the Sawyer robot from MetaWorld. We consider a Bang-Off-Bang policy encoded via a 3-bin Categorical distribution and use the same set of hyperparameters across all tasks. Performance mean and standard deviation across 10 seeds are provided in Figure 5 and we compare to the recent DMPO and D4PG agents, with additional training details provided in Appendix A. We note that DecQN exhibits very strong performance with respect to the continuous baselines and sometimes even improves upon them, despite operating over a heavily-constrained action subspace. This extends even to the high-dimensional Humanoid and Dog domains, with action dimensions $dim(\mathcal{A}) = 21$ and $dim(\mathcal{A}) = 38$, respectively. Significant research effort has produced highly capable continuous control algorithms through advances in representation learning, constrained optimization and targeted exploration. It is then surprising that a simple decoupled critic operating over a discretized action space and deployed with only an $\epsilon$-greedy policy is able to provide highly competitive performance.

### 5.2.1 SCALABILITY OF THE DECOUPLED REPRESENTATION

To put the benchmarking results into perspective, we compare DecQN to the non-decoupled DQN agent for a 3-bin discretization each. DQN conventionally enumerates the action space and scales exponentially in the action dimensionality, whereas DecQN only scales linearly (see also Figure 2). According to Figure 6, performance of DecQN and DQN remains comparable on low-dimensional tasks, however, for high-dimensional tasks the DQN agent quickly exceeds memory limits due to the

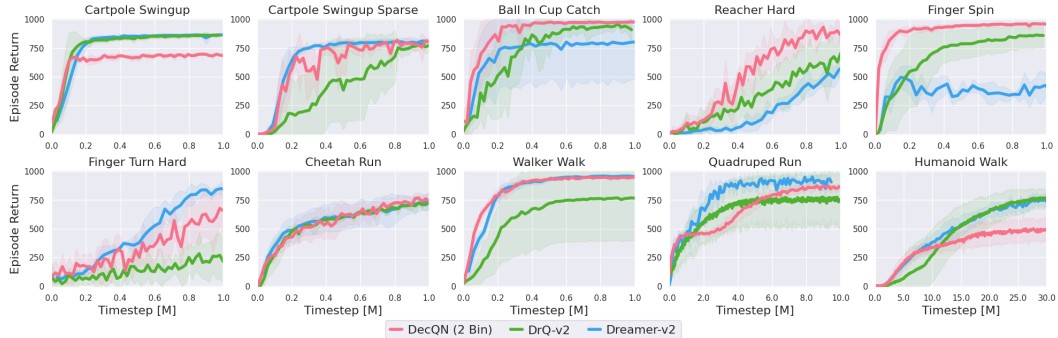

Figure 6: Comparison between DecQN and regular DQN on a selection of Control Suite tasks. Decoupling enables scaling to complex environments where DQN fails (dotted blue = memory error). DecQN's linear scaling further allows for efficient learning of fine-grained control (21 bins, green).

Figure 7: Performance on pixel-based control tasks from the DeepMind Control Suite, comparing DecQN with bang-bang policy to the continuous DrQ-v2 and Dreamer-v2 agents. We note that DecQN successfully accommodates the additional representation learning challenges. The best performing runs on Humanoid indicate that DecQN can efficiently solve complex tasks from vision, potentially requiring environment-specific hyperparameter settings or more sophisticated exploration.

large number of output parameters required. This contrast is amplified when significantly increasing granularity of DecQN's discretization to 21 bins. Generally, increasing granularity increases the action space and in turn yields more options to explore. While exploration over finer discretizations is likely to delay convergence, we observe highly competitive performance with both 3 and 21 bins.

## 5.3 BENCHMARKING: CONTINUOUS CONTROL FROM PIXEL INPUTS

We further evaluate on a selection of visual control tasks to see whether DecQN's decoupled discrete approach is compatible with learning latent representation from visual input. To this end, we combine DecQN with a convolutional encoder trained under shift-augmentations as proposed by Yarats et al. (2021). Figure 7 compares DecQN with a bang-bang parameterization (2-bin) to the state-of-the-art model-free DrQ-v2 and model-based Dreamer-v2 agents. DecQN and Dreamer-v2 employ a fixed set of hyperpa-

Table 1: Per-learning iteration runtime for Quadruped Run

| Agent | Runtime [s] |
|---|---|
| DecQN | $10.0 \pm 0.1$ |
| DrQ-v2 | $10.7 \pm 0.2$ |
| Dreamer-v2 | $29.0 \pm 0.8$ |

rameters across all tasks, while DrQ-v2 leverages a few task-specific variations. We note that DecQN is highly competitive across several environments while relinquishing some performance on the complex Humanoid task. Scaling DecQN to efficient learning of latent representation for high-dimensional systems under action penalties will be an interesting challenge. As the highest performing runs of DecQN on Humanoid Walk reach competitive performance, we are optimistic that this will be viable under potential parameter adjustments or more sophisticated exploration strategies. As both DrQ-v2 and Dreamer-v2 provide limited data on Quadruped Run (3M and 2M timesteps, respectively), we further re-ran the authors' code on our hardware and record average timings for one cycle of data collection and training. We observe favorable runtime results for DecQN in Table 1 and estimate that code optimization could further reduce timings as observed for DrQ-v2.

## 5.4 COMMAND-CONDITIONED LOCOMOTION CONTROL

We additionally provide qualitative results for learning a command-conditioned locomotion controller on a simulated Mini Cheetah quadruped to demonstrate the broad applicability of decoupled

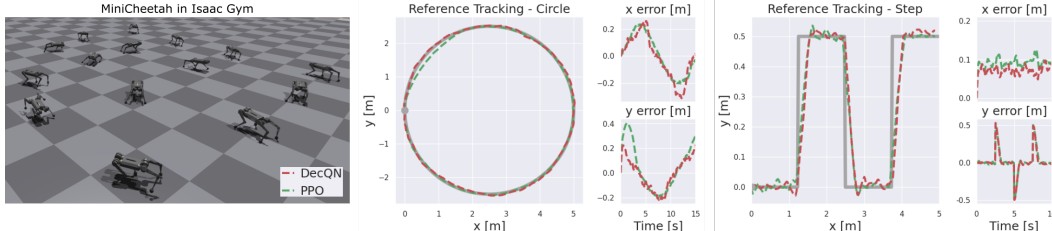

Figure 8: Qualitative results for learning a command-conditioned locomotion controller on a simulated Mini Cheetah. The agent learns to predict position targets for joint-level PD controllers with the objective of tracking commanded base velocities. We observe good tracking performance for following references in state-space at the level of PPO, which is a common choice in these settings.

discretized control. We consider a minimal implementation of DecQN in PyTorch simulated with Isaac Gym. The agent aims to track velocity commands by predicting position targets for the leg joints, which are then provided to a low-level proportional-derivative (PD) controller. As targets are generated in joint angle space, we consider a 31-bin discretization. We evaluate the resulting command-conditioned policy based on tracking performance when following a reference position trajectory. To this end, velocity commands are generated by a simple proportional controller on the position error between the reference and the robot base. Figure 8 showcases tracking for two references. The circular trajectory has a diameter of $5\,\mathrm{m}$ and is tracked at a velocity of $1\,\mathrm{m/s}$ (middle), while the step trajectory introduces lateral jumps of $0.5\,\mathrm{m}$ and is tracked at a velocity of $0.7\,\mathrm{m/s}$ (right). In both scenarios we observe good tracking performance at levels competitive with the commonly used PPO agent (Rudin et al., 2022). The system reacts well even to strong jumps in the reference with quick recovery in response to slight overshooting. This underlines the versatility of simple decoupled discrete control, learning policies for various control types including application of acceleration or velocity control and generating position targets for downstream control systems.

## 6 CONCLUSION

Recent successes in data-efficient learning for continuous control have been largely driven by actor-critic algorithms leveraging advances in representation learning, constrained optimization, and targeted exploration. In this paper, we take a step back and investigate how a simple variation of Q-learning with bang-bang action discretization and $\epsilon$-greedy exploration compares to current state-of-the-art algorithms on a variety of continuous control tasks. Decoupled Q-Networks (DecQN) discretizes the action space, fully decouples individual action dimensions, and frames the single-agent control problem as a cooperative multi-agent optimization. This is then solved via centralized training with decentralized execution, where the decoupled critic can be interpreted as a discrete analog to the commonly used continuous diagonal Gaussian policies. We observe highly competitive performance of DecQN on state-based control tasks while comparing to the state-of-the-art D4PG and DMPO agents, extend the approach to pixel-based control while comparing to the recent DrQ-v2 and Dreamer-v2 agents, and demonstrate that the method solves more complex command-conditioned tracking control on a simulated Mini Cheetah quadruped.

DecQN is highly versatile, learning capable policies across a variety of tasks, input and output modalities. This is intriguing, as DecQN does not rely on many of the ingredients associated with state-of-the-art performance in continuous control and completely decouples optimisation and exploration across the agent's action dimensions. We provide intuition for how the decoupled critic handles coordination challenges by studying variations of matrix games proposed in prior work. We observe that shared access to the global observations facilitates coordination among decoupled actuators. Furthermore, it is often sufficient for a critic to accurately reflect relative decision quality under the current state distribution to find the optimal policy. Promising future work includes the interplay with guided exploration and the investigation of these concepts on physical systems.

There is great value in contrasting complex state-of-the-art algorithms with simpler ones to dissect the importance of individual algorithmic contributions. Identifying the key ingredients necessary to formulate competitive methods aides in improving the efficiency of existing approaches, while providing better understanding and foundations for future algorithm development. To this end, the DecQN agent may serve as a straightforward and easy to implement baseline that constitutes a strong lower bound for more sophisticated methods, while raising interesting questions regarding how to assess and compare continuous control benchmark performance.

ACKNOWLEDGMENTS

Tim Seyde, Peter Werner, Wilko Schwarting, Igor Gilitschenski, and Daniela Rus were supported in part by the Office of Naval Research (ONR) Grant N00014-18-1-2830 and Qualcomm. This article solely reflects the opinions and conclusions of its authors and not any other entity. We thank them for their support. The authors further would like to thank Danijar Hafner for providing benchmark results of the Dreamer-v2 agent, and acknowledge the MIT SuperCloud and Lincoln Laboratory Supercomputing Center for providing HPC resources.

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

# A   TRAINING DETAILS

Experiments on Control Suite and Matrix Game tasks were conducted on a single NVIDIA V100 GPU with 4 CPU cores (state-based) or 20 CPU cores (pixel-based). Experiments in MetaWorld and Isaac Gym were conducted on a single NVIDIA 2080Ti with 4 CPU cores. Benchmark performance of DecQN is reported in terms of the mean and one standard deviation around the mean on 10 seeds. We implemented DecQN within the Acme framework (Hoffman et al., 2020) in TensorFlow. For the Mini Cheetah (Katz et al., 2019) task we implement a version of DecQN in PyTorch.

**Hyperparameters**   We provide hyperparameter values of DecQN used for benchmarking in Table 2. A constant set of hyperparameters is used throughout all experiments, with modifications to the network architecture for vision-based tasks. For the matrix games, we further set the $n$-step parameter to 1 to account for the underlying timescale and direct impact that actions have on both the state transition and reward. Results for DQN were obtained with the same parameters without decoupling. The baseline algorithms used the default parameter settings provided in the official implementations by the original authors, within the Acme library in the case of D4PG and DMPO.

Table 2: DecQN hyperparameters for state- and pixel-based control.

| Parameter | Value [State] | Value [Pixel] |
|---|---|---|
| Optimizer | Adam | Adam |
| Learning rate | $1 \times 10^{-4}$ | $1 \times 10^{-4}$ |
| Replay size | $1 \times 10^{6}$ | $1 \times 10^{6}$ |
| $n$-step returns | 3 | 3 |
| Action repeat | 1 | 1 |
| Discount $\gamma$ | 0.99 | 0.99 |
| Batch size | 256 | 256 |
| Hidden size | 500 | 1024 |
| Bottleneck size | — | 100 |
| Gradient clipping | 40 | 40 |
| Target update period | 100 | 100 |
| Imp. sampling exponent | 0.2 | 0.2 |
| Priority exponent | 0.6 | 0.6 |
| Exploration $\epsilon$ | 0.1 | 0.1 |

**Architecture**   For the state-based experiments, we leverage a fully-connected architecture that includes a single residual block followed by a layer norm. For the vision-based experiments, we leverage the convolutional encoder with bottleneck layer from Yarats et al. (2021) followed by two fully-connected layers. In both cases, a fully-connected output layer predicts the decoupled state-action values so that the torso up to the final layer remains shared among the decoupled critics.

**Action discretization**   The continuous action space is discretized along each action dimension into evenly spaced discrete actions including the boundary values. Assuming symmetric action bounds, for axis $i$ this yields bang-bang control in the case of 2 bins with $\mathcal{A}_{n_{\mathrm{bin}=2},i} = \{-a_{\mathrm{bound},i}, +a_{\mathrm{bound},i}\}$ and bang-off-bang control in the case of 3 bins with $\mathcal{A}_{n_{\mathrm{bin}=3},i} = \{-a_{\mathrm{bound},i}, 0, +a_{\mathrm{bound},i}\}$, and so on.

**Decoupling**   The decoupling based on value decomposition (Sunehag et al., 2017) only requires small modifications of the original DQN structure (see also Sharma et al. (2017)). The output layer size is adapted to predict all state-action utilities for action dimensions $n_a$ and discrete bins $n_b$ to yield output size $[..., n_a n_b]$, where $[...]$ indicates the batch dimensions. This is reshaped into $[..., n_a, n_b]$, where the combined state-action value is computed either by indexing with the observed bin at the current state or maximizing over bins at the next state, followed by a mean over action dimensions to recover state-action values.

**Licenses**   The Acme library is distributed under the Apache-2.0 license and available on GitHub. The D4PG and DMPO agents are part of the Acme library, while both DecQN and DQN for continuous control were implemented within the Acme framework. Both Dreamer-v2 and DrQ-v2 are

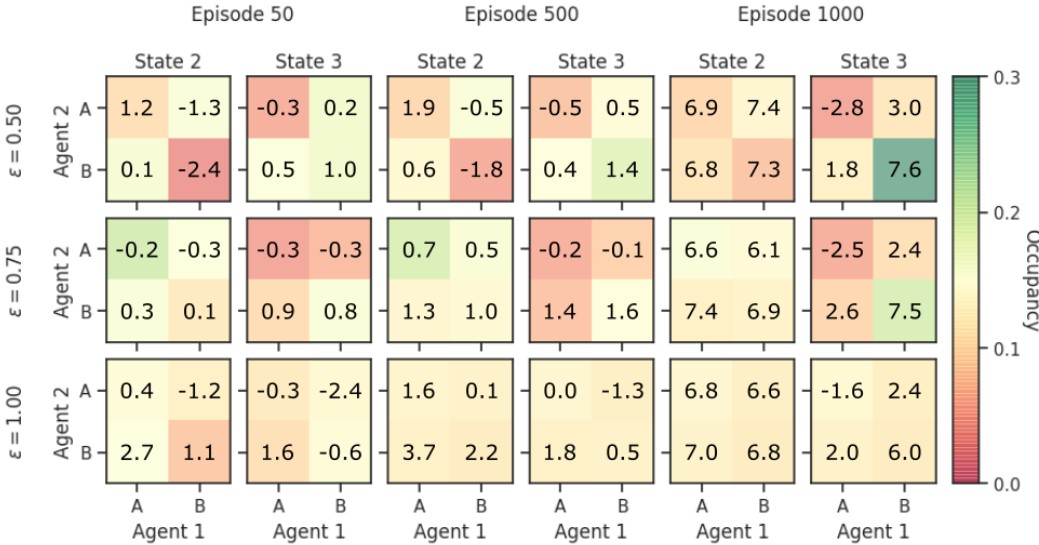

Figure 9: State distribution and action-values for DecQN on the two-step game from Section 5.1. We consider 3 training stages pre-convergence across 5 seeds. Color indicates cumulative state-action occurrence, numbers represent predicted mean action values. While the decoupled agent struggles to represent action values under a uniform distribution ($\epsilon = 1.0$, bottom), it accurately represents values under the current policy when directly influencing the state distribution ($\epsilon = 0.5$, top).

distributed under the MIT license and are available with their benchmarking results on GitHub here and here, respectively.

## B  STATE-ACTION DISTRIBUTION: TWO-STEP GAME

Expanding on the discussion of the two-step matrix game in Section 5.1 we provide state-action distribution data for the DecQN agent in Figure 9. We evaluate three stages of early training before convergence for 5 seeds. Color of state-action pairings indicates cumulative frequency within replay memory, while numbers represent mean predicted state-action values. We consider different degrees of randomness in the policy by varying the $\epsilon$ parameter. It can be noted that the agent fails to accurately represent the optimal state-action value in state 3 when learning based on a uniform state-action distribution (bottom row). However, enabling the agent to directly influence the underlying distribution through its policy results in accurate learning of state-action values around this policy (top and middle rows). While the decoupled agent may not be able to accurately reflect the full state-action value function, it can be sufficient to do so over a subspace relevant for solving the task.

## C  ENVIRONMENTS

A brief overview of the environments evaluated throughout this paper is provide in Table 3, grouped based on their distributors. In particular, we highlight the dimensionality of the state space $\mathcal{S}$ and action space $\mathcal{A}$ as well as the total number of timesteps used for training on each environment. For the Mini Cheetah task, we note that the number of steps - marked with an $*$ - denotes the approximate number of steps per environment, where we use $1024$ parallel environments within Isaac Gym.

## D  ADDITIONAL BASELINES

We provide additional baselines on a selection of Control Suite environments in Figure 10. The baselines consist of a continuous-control algorithm with Categorical head based on MPO from Seyde et al. (2021), the continuous-control SAC agent from Yarats & Kostrikov (2020), as well as the critic-only methods QT-Opt and AQL-Seq based on data from Van de Wiele et al. (2020). DecQN is

Table 3: Description of benchmark environments used throughout the paper.

| Suite | Task | $dim(\mathcal{S})$ | $dim(\mathcal{A})$ | Steps [State] | Steps [Pixel] |
|---|---|---|---|---|---|
| Control Suite | Ball in Cup Catch | 8 | 2 | — | $1 \times 10^6$ |
| | Cartpole Swingup | 4 | 1 | $1 \times 10^6$ | $1 \times 10^6$ |
| | Cartpole Swingup Sparse | 4 | 1 | $2 \times 10^6$ | $1 \times 10^6$ |
| | Cheetah Run | 18 | 6 | $3 \times 10^6$ | $1 \times 10^6$ |
| | Dog Run | 158 | 38 | $5 \times 10^6$ | — |
| | Dog Trot | 158 | 38 | $5 \times 10^6$ | — |
| | Dog Walk | 158 | 38 | $5 \times 10^6$ | — |
| | Finger Spin | 6 | 2 | $2 \times 10^6$ | $1 \times 10^6$ |
| | Finger Turn Hard | 6 | 2 | $2 \times 10^6$ | $1 \times 10^6$ |
| | Humanoid Run | 54 | 21 | $30 \times 10^6$ | — |
| | Humanoid Stand | 54 | 21 | $10 \times 10^6$ | — |
| | Humanoid Walk | 54 | 21 | $20 \times 10^6$ | $30 \times 10^6$ |
| | Quadruped Run | 56 | 12 | $5 \times 10^6$ | $10 \times 10^6$ |
| | Reacher Hard | 4 | 2 | $2 \times 10^6$ | $1 \times 10^6$ |
| | Walker Run | 18 | 6 | $3 \times 10^6$ | — |
| | Walker Walk | 18 | 6 | $1 \times 10^6$ | $1 \times 10^6$ |
| Isaac Gym | Mini Cheetah Tracking | 48 | 12 | $3 \times 10^{6*}$ | — |
| Matrix Games | Two Step | 3 | 4 | $6 \times 10^4$ | — |
| | Penalty | 4 | 9 | $2 \times 10^5$ | — |
| | Climbing | 4 | 9 | $2 \times 10^5$ | — |
| Meta World | Assembly | 39 | 4 | $2 \times 10^6$ | — |
| | Door Open | 39 | 4 | $2 \times 10^6$ | — |
| | Drawer Open | 39 | 4 | $2 \times 10^6$ | — |
| | Hammer | 39 | 4 | $2 \times 10^6$ | — |
| | Pick Place | 39 | 4 | $2 \times 10^6$ | — |

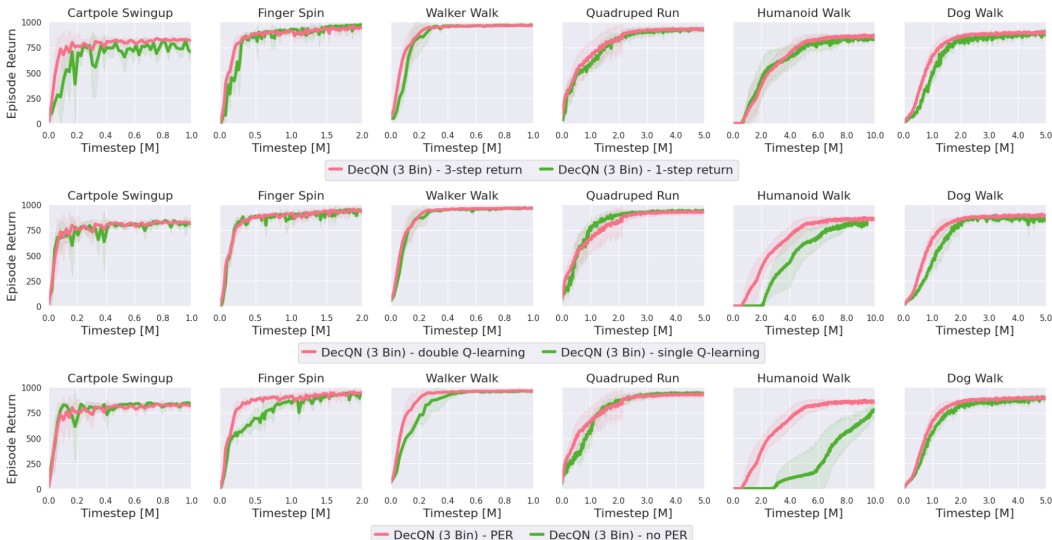

Figure 10: Comparison of DecQN to MPO with Categorical policy head, continuous SAC, as well as critic-only QT-Opt and AQL-Seq. DecQN displays state-of-the-art performance without relying on actor-critic methods or continuous control, while remaining sample-efficient in high-dimensional action spaces by avoiding sampling-based methods.

Figure 11: Ablations of the DecQN agent on multi-step returns, double Q-learning, and prioritized experience replay (top to bottom). The agent is robust to changes in individual components while profiting from PER to select useful interactions on complex multi-phase tasks such as the Humanoid.

competitive with discretized MPO, further underlining that actor-critic methods are not required to obtain strong benchmark performance. The additional SAC baseline shows improved performance on Humanoid at the cost of being unable to learn on the Dog task, mirroring results of Hansen et al. (2022). Generally, we believe that most current state-of-the-art continuous control algorithms are capable of achieving comparable benchmark performance. However, these results do not appear to be conditional on using continuous control or actor-critic methods, and can be achieved with much simpler Q-learning over discretized bang-bang action spaces with constant $\epsilon$-greedy exploration. We further provide converged performance of QT-Opt and AQL-Seq. While these methods remain competitive on low-dimensional tasks, their performance quickly deteriorates for high-dimensional action spaces due to their reliance on sampling this space.

# E    ABLATIONS ON RAINBOW COMPONENTS

We provide ablations on three components of the underlying Rainbow agent, namely multi-step returns, double Q-learning, and prioritized experience replay in Figure 11 (rows, respectively). Learning is generally robust to removal of individual components in light of cumulative reward at convergence, improving learning speed primarily on the more complex tasks. In particular, double Q-learning and PER provide a boost on the Humanoid task to enable state-of-the-art performance.

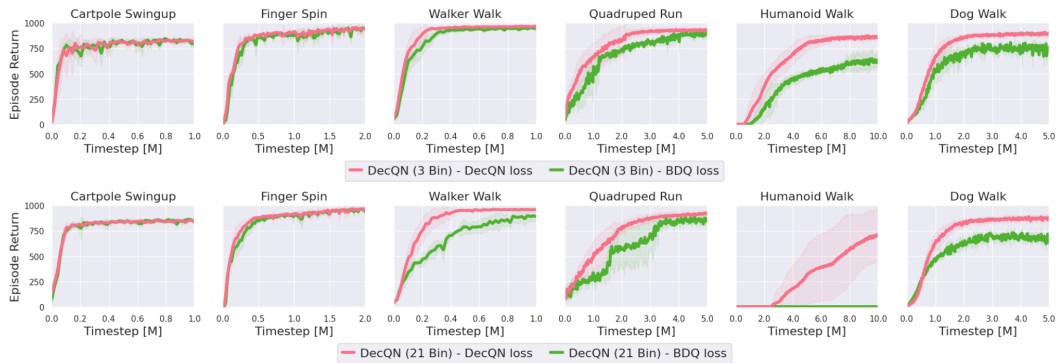

Figure 12: Comparison of DecQN with the original and BDQ loss for 3 (top) and 21 bins (bottom). Aggregating per-dimension utilities transforms independent into joint learners, yielding significant improvements in final performance and learning stability particularly for high-dimensional tasks.

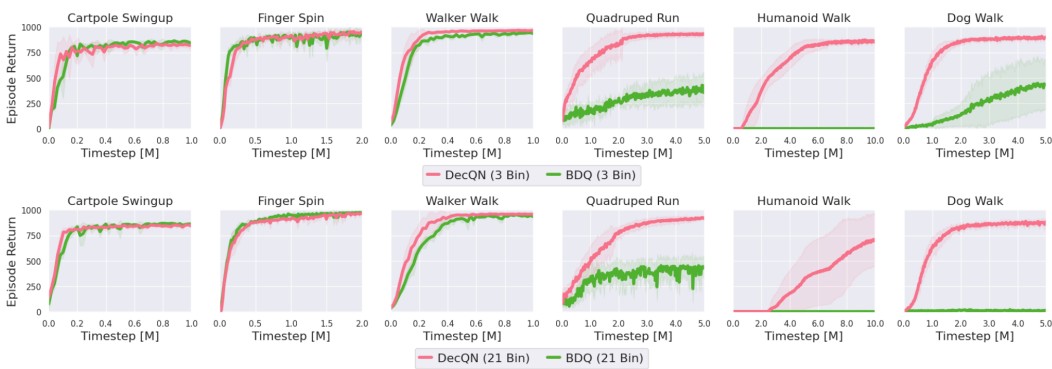

Figure 13: Comparison of DecQN and BDQ for 3 (top) and 21 bins (bottom). DecQN provides significant advantage on the more complex domains, underlining the importance of design choices such as strong architectural centralization and value decomposition via single-action utility functions.

## F   DISCUSSION OF BDQ AND HGQN

In the following, we provide a more detailed discussion of the relation to Branching Dueling Q-Networks (BDQ) Tavakoli et al. (2018) as well as Hypergraph Q-Networks (HGQN) Tavakoli et al. (2021) together with associated ablations. Generally, our motivation is to highlight that minimal changes to the original DQN agent enable state-of-the-art performance on continuous control benchmarks solely based on bang-bang Q-learning with constant exploration.

The BDQ agent from Tavakoli et al. (2018) considers independent learning of per-dimension state action values, employing a dueling architecture with separate branches for each action dimension and exploration based on a Gaussian with scheduled noise in combination with fine-grained discretizations. DecQN forces a higher degree of centralization by predicting per-dimension state-action utilities without intermediate branching or dueling heads for joint learning within a value decomposition, while using constant $\epsilon$-greedy exploration with a focus on only coarse bang-bang control. The most important difference is independent learning in comparison to joint learning based on value decomposition. We provide an ablation of our approach that does not aggregate per-action dimension values and thereby mimics BDQ's independent learning in Figure 12. Particularly for high-dimensional tasks learning a value decomposition can significantly improve performance, an effect that is amplified when selecting more fine-grained discretizations. We furthermore evaluated the original BDQ agent after modifying the default parameters (i.e. batch size, target update frequency, learning frequency, gradient clipping, exploration strategy, multi-step returns). The results provided in Figure 13 were obtained based on the best configuration we found by increasing the batch size and treating each episode as infinite-horizon. While we believe that the results in

Figure 14: Comparison of DecQN with original and HGQN aggregation. Aggregating via the mean induces a more favorable scaling of the reward in the TD-error, which we found to enable graceful scaling to complex task with high-dimensional action spaces to achieve state-of-the-art performance.

Figure 12 offer a more informative comparison, we observe a similar trend regarding scaling to high-dimensional tasks in both cases.

The hypergraph Q-networks (HGQN) framework from Tavakoli et al. (2021) considers value decomposition across subsets of action dimensions and introduces higher-order hyperedges between action groupings. DecQN can therefore be interpreted as an instance of the conceptual HGQN (r=1) formulation that leverages single-action decomposition without higher-order edges as had previously been investigated with the Atari-based FARAQL agent (Sharma et al., 2017). Our primary focus is on simplicity and showing how far we can push basic concepts that constitute capable alternatives to more sophisticated recent algorithms. There are further several differences in per-dimension utility aggregation, architectural choices regarding the use of branching, loss function, exploration scheduling, as well as in the usage of PER and double Q-learning for continuous control. We provide a brief ablation on using the mean compared to sum aggregation when computing Bellman targets in Figure 14. While this appears as a subtle difference, we found that leveraging the mean yields graceful scaling to complex tasks with high-dimensional action spaces without any parameter adjustments. This enables state-of-the-art performance across a wide range of environments and input-output modalities with a single set of hyper-parameters.

Generally, our motivation is not to advocate for a novel algorithm that should be the go-to method for solving continuous control problems. Our core objective is to highlight that current state-of-the-art continuous control benchmark performance is at the level of decoupled Q-learning over bang-bang parameterized action spaces with constant exploration. This requires only minor if well-directed modification to the original DQN algorithm and yields strong performance for both feature- and pixel-based observation spaces as well as acceleration-, velocity-, and position-based action spaces. Our investigation provides additional motivation for existing work while establishing closer connections to classical MARL coordination challenges, as well as extensive experimental evaluation in comparison to current state-of-the-art algorithms.

## G   RAINBOW DQN AGENT

We leverage several modifications of vanilla DQN that accelerate learning and improve stability based on the Rainbow implementation provided by Acme (Hessel et al., 2018; Hoffman et al., 2020):

**Target Network**   Bootstrapping directly from the learned value function can lead to instabilities. Instead, evaluating actions based on a target network $Q_{\theta^-}(s_t, a_t)$ improves learning (Mnih et al., 2015). The target network's weights $\theta^-$ are updated periodically to match the online weights $\theta$.

**Double Q-learning**   Direct maximization based on the value target can yield overestimation error. Decoupling action selection from action evaluation improves stability (Van Hasselt et al., 2016). Action selection then queries the online network, while action evaluation queries the target network. Our implementation further leverages two sets of critics, where we bootstrap based on their average during learning and take their maximum during action selection.

**Prioritized Experience Replay**   Uniform sampling from replay memory limits learning efficiency. Biasing sampling towards more informative transitions can accelerate learning (Schaul et al., 2015). The observed temporal difference error can serve as a proxy for expected future information content.

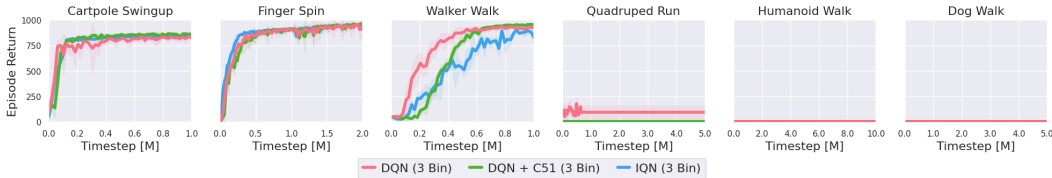

Figure 15: Comparison of DecQN, DecQN with a distributional C51 critic, and the distributional IQN agent. The distributional version of DecQN generally yields slightly improved performance at convergence, while decoupling improves performance over the non-decoupled IQN baseline.

Figure 16: Comparison of DQN, DQN with a distributional C51 critic, and the distributional IQN agent. Without decoupling, we do not observe benefits of distributional critics in these domains. Slower convergence on Walker Walk with distributional representations could indicate that the associated increased parameter count translates to a more difficult optimization problem.

**Multi-step Returns**   Instead of directly bootstrapping from the value function at the next state, evaluating the reward explicitly over several transitions as part of a multi-step return, such that $G_{t:t+n} = R_t + \cdots + \gamma^{n-1}R_{t+n-1} + \gamma^n V_{t+n}(S_{t+n})$ can improve learning (Sutton & Barto, 2018).

## H   CONTROL-AFFINE DYNAMICS AND LINEAR REWARDS

Under deterministic environment dynamics and policy we can simplify the Bellman equation as

$$Q^\pi(s_t, a_t) = r(s_t, a_t) + \gamma V^\pi(s_{t+1}). \tag{5}$$

Consider a reward structure that is linear in the action dimensions with $r(s_t, a_t) = \sum_{j=0}^{M} r^j(s_t, a_t^j)$. Under given transition tuples and fixed target values, the TD(0) objective can then be formulated as

$$Q^\pi(s_t, a_t) = \sum_{j=0}^{M} \left( r^j(s_t, a_t^j) + \frac{\gamma}{M} V^\pi(s_{t+1}) \right), \tag{6}$$

which can be solved exactly based on a linear value decomposition $Q(s_t, a_t) = \sum_{j=0}^{M} Q_j(s_t, a_t^j)$. Particularly for robotics applications, many common reward structures depend (approximately) linearly on the next state observation while the system dynamics are control-affine such that $s_{t+1} = f(s_t) + g(s_t)a_t$, with $f(s_t)$ and $g(s_t)$ only depending on the current state $s_t$. While these are strong assumptions, it may provide intuition for why the problem structures considered here may be amenable to decoupled local optimization and for the observed highly competitive performance.

## I   DISTRIBUTIONAL CRITICS

We briefly investigate replacing our deterministic critic with a distributional C51 head (Bellemare et al., 2017) The decomposition now proceeds at the probability level via logits $l = \sum_{j=1}^{M} l_j / M$ during the C51 distribution matching. We do not make any parameter adjustments and compare performance with and without a distributional critic for both DecQN and DQN in Figures 15 and 16, respectively, and provide the IQN agent as an extension of the QR-DQN agent for reference (Dabney et al., 2018a;b). Our empirical evaluation suggests that a distributional critic can slightly increase performance at convergence and sample-efficiency. Both DecQN and DecQN + C51 yield similar performance on average with some environment specific variations. Without decoupling, DQN

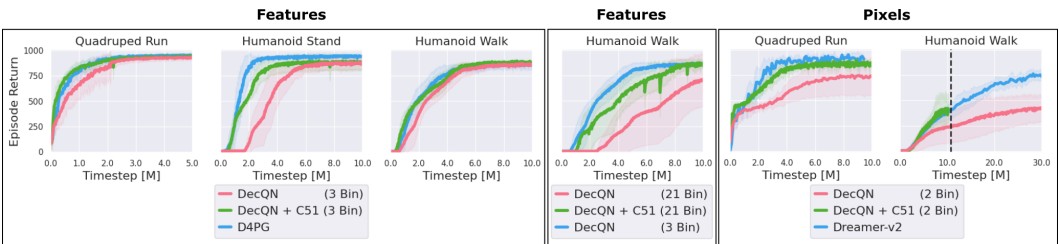

Figure 17: Comparison of the distributional versions of DecQN, DQN, and IQN. Decoupling of the value representation in conjunction with bang-bang action representations plays a key role in scaling these approaches to high-dimensional continuous control tasks.

Figure 18: DecQN + C51 on tasks where regular DecQN did not perform at least as good as the best baseline. Adding a distributional critic can further boost performance, underlining the strength and versatility of this very simple approach (vertical line = current training status due to time constraints).

yields slightly faster convergence than DQN + C51 on Walker Walk which could indicate that the increased parameter count of distributional critics translates to a more challenging optimization problem (see also the immediate memory error of non-decoupled distributional agents on Quadruped Run in Figure 16.) Both DecQN and DQN with or without distributional critic improve performance over the IQN baseline. We further provide performance of only the distributional agents in Figure 17 for ease of comparison. It is likely that given sufficient tuning the performance of our distributional variations could be increased even further, while we note that the deterministic DecQN agent already matches performance of the distributional D4PG and DMPO actor-critic agents. A more extensive investigation into decoupled distributional Q-learning provides promising avenues for future work.

We note that without any parameter adjustments or tuning, replacing the deterministic critic with the distributional C51 critic in DecQN significantly boosts performance on the few environments where DecQN did not perform at least as good as the best baseline. This applies to large bin sizes, feature inputs and pixel inputs as exemplified in Figure 18 (vertical line in Humanoid Walk from pixels denotes current training status, subject to time constraints). These results further underline the versatility and strength of this very simple approach.

## J STOCHASTIC ENVIRONMENTS

We extend our study to stochastic versions of a selection of Control Suite tasks. The results in Section 5.1 indicate that coordination among decoupled actors becomes more difficult if the action selection of other actors is less stationary from the perspective of each individual actor. To increase stochasticity, we consider both observation and reward noise represented by additive Gaussian white noise with standard deviation $\sigma_{\text{noise}} = 0.1$. Figures 19 and 20 provide results for DecQN, D4PG, and the distributional DecQN + C51 under observation and reward noise, respectively. While we observe similar performance of DecQN and D4PG on most tasks, performance of DecQN is visibly reduced on Humanoid Walk. Humanoid Walk combines several aspects that can hinder exploration, including implicit staged reward (first get up, then walk) and action penalties, which likely exacerbate coordination challenges when combined with stochasticity. We therefore believe that special care should be taken when applying regular DecQN in environments that have the potential to amplify the coordination challenges observed in the simple matrix game domains of Section 5.1. Adding a distributional critic to DecQN allows for better modelling of stochasticity and visibly improves

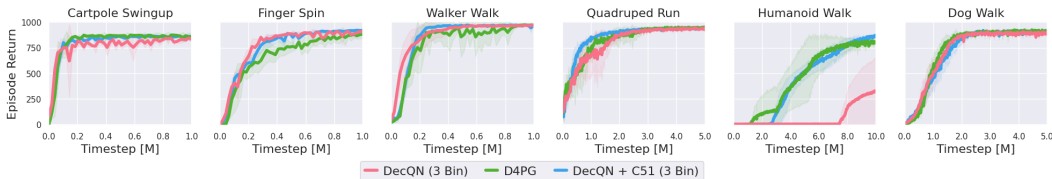

Figure 19: Stochastic observation tasks adding Gaussian white noise to the observations ($\sigma_{\text{noise}} = 0.1$). DecQN appears less robust to observation noise than D4PG, mirroring the findings regarding coordination challenges under high stochasticity in matrix games of Section 5.1. Adding a distributional critic improves performance and yields faster convergence than D4PG on Quadruped Run.

Figure 20: Stochastic reward tasks adding Gaussian white noise to the rewards ($\sigma_{\text{noise}} = 0.1$). DecQN is less robust to reward noise than the distributional D4PG, mirroring the findings regarding coordination challenges under high stochasticity in matrix games of Section 5.1. With a distributional critic, DecQN can directly account for stochastic returns and matches or outperforms D4PG.

performance in stochastic environments, where DecQN + C51 even improves on the distributional D4PG agent in some environments.

