# OpenReview forum: "Solving Continuous Control via Q-learning"
_ICLR.cc/2023/Conference — ICLR 2023 poster_

### Official Review · Reviewer_zBqq · 2022-10-21

**Confidence:** 5
**Correctness:** 3
**Technical Novelty And Significance:** 3
**Empirical Novelty And Significance:** 3
**Recommendation:** 8

**Clarity, Quality, Novelty And Reproducibility:**

I am happy with the quality of the experiments, and I believe the results will be easily reproducible. The baseline performances also match what I would expect (based on comparing them with published results).

The method is not original, but I like the work in that it is the first large-scale evaluation of this super-simple method and against strong baselines (including model-based and model-free). I highly believe this paper could help bring more interest to solving continuous control problems via discretization and Q-learning.

The exposition of the ideas is clear, and the paper cites all relevant literature. Nonetheless, considering the nature of the contribution is mostly in the large-scale evaluation and not the proposed methodology/paradigm, I believe the tone in the abstract and also the intro regarding contributions should be softened.

**Strength And Weaknesses:**

**Strengths:**
- Exploring critic-only methods (e.g. Q-Learning) for continuous control problems is interesting because: (1) they are much simpler; (2) they allow us to have a unified approach for tackling discrete and continuous action problems; ...

- Using discretization for tackling continuous control problems is interesting because: (1) because the agent can potentially capture non-parametric multimodal policy distributions in actor-critic methods; (2) many advances for discrete-action Q-learning (for exploration and credit assignment) can be immediately adopted; (3) curriculum learning can be achieved by growing the action space gradually; ...

- The experiments are done on a vast and appropriate set of continuous control benchmarks.

- The baselines capture a good range of SOTA model-free and model-based methods for the chosen benchmarks.

- Exposition of the connection of the method to multi-agent RL is appropriate and useful.


**Weaknesses:**

- The approach (i.e. Decoupled Q-Learning) is not novel: it is similar to HGQN (r=1) from (Tavakoli et al., 2021) (which the authors have referenced), but where the authors replace the **sum** mixing function with a **mean** mixer to achieve less sensitivity to learning rates in domains with varying action dimensionality. This is an appropriate choice but nonetheless one that is supported under the Action Hypergraph Networks framework.

- Due to the above reason, I'm not a fan of the completely new name for the method in this paper - but it is not a major issue.

- The abstract makes it seem like this paper is proposing or achieving the following for the first time:

   - This paper is not the first to solve and achieve competitive performance wrt. continuous-action actor-critic methods in continuous control benchmarks via discretization and a scalable version of Q-learning (e.g. Metz et al. (2017) and Tavakoli et al. (2018) are early approaches that have done so).

   - This paper is not the first to draw a connection between discrete multi-dimensional-action-space RL and multi-agent RL to achieve scalability wrt. increasing action dimensionality (e.g. Tavakoli et al. (2018) is an early instance in the context of Deep RL).

   - This paper is not the first to exploit value-decomposition within Q-learning in single-agent RL (Sharma et al. (2017) and Tavakoli et al. (2021) are two instances).

**Summary Of The Paper:**

This paper focuses on evaluating the power of a critic-only approach to solving continuous control problems, by combining Q-learning with value decomposition (an approach from multi-agent RL under the *centralized training but decentralized execution* paradigm). They test this approach across a large set of environments from continuous control benchmarks and show that this simple model-free approach performs on par with SOTA actor-critic methods, both for model-free and model-based methods, and based on state features and pixel observations.

**Summary Of The Review:**

Great empirical evaluation of a simple discrete-action critic-only model-free method on a large set of continuous control benchmarks (including some complex ones) against much more involved SOTA model-free and model-free actor-critic methods for continuous control; showing competitive performance. The method is not novel, but the study is interesting and insightful and the paper should be accepted in my view due to the potential impact it can have: showing strong evidence that simple Q-learning is sufficient for solving high-dimensional continuous control problems, via discretization and combining insights from multi-agent RL to achieve scalability.

My only concern is overclaiming contributions in the abstract and intro. The authors appropriately cite all related works, to the best of my knowledge, but should tone down the claims particularly in the abstract.

---

> ### Author Response · Authors · 2022-11-15
> **Response to Reviewer zBqq**
>
> Thank you very much for your positive comments and for appreciating our extensive empirical evaluation. We are excited to present **additional experiments** focusing on distributional RL and stochastic environments.
>
> We have updated the paper to include these experiments in Appendices I & J (changes in blue) and adapted part of the abstract and introduction based on your suggestions. We agree that our work builds on and extends excellent prior work, providing additional motivation while focusing on simplicity and investigating the minimal requirements to achieve SOTA performance across a breadth of benchmarks.
>
> We are very excited about new experiments that extend our study to distributional RL in Appendix I & J. Our empirical results indicate that decoupling pairs well with distributional critics (Appendix I), and is particularly valuable in stochastic environments that can induce failure cases in non-distributional critics (Appendix J). We also believe that more extensive investigations into decoupled distributional Q-learning provides promising avenues for future work.
>
> Thank you very much again for your positive feedback, and we wholeheartedly agree about the value of further increasing excitement for critic-only methods in the community.

---

> > ### Comment · Reviewer_zBqq · 2022-11-16
> > **Thanks!**
> >
> > Thanks for your response and adjustments.
> >
> > Could you comment on why stochasticity in Dog (with 38 action dimensions) is less detrimental to DecQN than in Humanoid (with 21 action dimensions).
> >
> > **Minor:**
> >
> > - Caption of Fig. 15: I think the last sentence is slightly odd. Perhaps you forgot an adjective, e.g. *significantly improves performance* in the second half.

---

> > > ### Author Response · Authors · 2022-11-17
> > > **Stochastic environments**
> > >
> > > That's a great point!
> > >
> > > From our experience, the Humanoid task combines some features that make it particularly challenging. The reward is implicitly staged by requiring first a get-up motion with stabilization, followed by walking. If the agent is unable to find and coordinate the get-up behavior among noisy reward signals it will remain on the ground with near-zero reward. These challenges are amplified by action penalties (limiting exploration) and inherent instability of bipedal walking. The Dog tasks start the agent in more favorable configurations with increased stability of quadrupedal gait and no action penalties. We therefore believe that the stronger impact of added stochasticity on the Humanoid is primarily due to amplification of these coordination challenges during the initial exploration of the get-up motion.

---

> > > > ### Comment · Reviewer_zBqq · 2022-11-17
> > > > **Makes sense**
> > > >
> > > > Thanks for providing a potential explanation. It makes sense. Would be nice to briefly clarify this in the paper to avoid confusion. But this can be done for the final version. Thanks again.

---

> > > > > ### Author Response · Authors · 2022-11-19
> > > > > **Thanks!**
> > > > >
> > > > > Sounds great, we will add the above description of Humanoid vs. Dog task characteristics to the final version - thanks for bringing it up!

---

### Official Review · Reviewer_FMD4 · 2022-10-24

**Confidence:** 4
**Correctness:** 4
**Technical Novelty And Significance:** 1
**Empirical Novelty And Significance:** 2
**Recommendation:** 5

**Clarity, Quality, Novelty And Reproducibility:**

I have concerns regarding novelty and originality of the work.

Suppose that I had no concern regarding that parameters used for training (such as in foundational models). Then, how would this method compare against an ensemble of Q functions. There is also prior work in distributional RL such as QR-DQN, which approximates the distributional Q-value with multiple Q-functions. This would be better to benchmark against, as opposed to vanilla DQN.



**Strength And Weaknesses:**

Strengths:
- The authors intelligently framed the solution as cooperative MARL.
- Evaluation in Section 5.1 is strong and shows that  DecDQN can coordinate across Q-networks.
- 10 seeds per agent

Weaknesses:
- The baselines are not up to date with other RL agents, such as SAC and DDPG. Would there be similar observed benefits with different algorithms and replacing the Q-networks with the paper's proposed formulation?
- Figure 6, why does 21 bin Dec DQN generally do worse than 3 bin?
- Is Sec 5.3 necessary? Dreamer-v2 would be slower due to being a model-based approach which looks ahead.
-  (Nit) want to cite the library used to train other baselines (assuming it is RLlib)
- Is there a failure case where DecDQN would fail when compared to other baselines?

**Summary Of The Paper:**

The paper attempts to solve the problem of higher dimensional actions spaces in the context of deep reinforcement learning. The authors propose DecDQN, which consists of multiple Q-networks over different sets of the action space, resulting in reduced network parameters and potentially better performance.

**Summary Of The Review:**

Overall, the paper is well written and clear. The results are logical and structured. However, there are concerns regarding novelty of the idea. I am willing to improve the score as long as the authors better address methods in distributional RL (such as QR-DQN) and add it to the results.

---

> ### Author Response · Authors · 2022-11-15
> **Response to Reviewer FMD4 (1 / 2)**
>
> Thank you very much for your feedback and for highlighting our strong empirical evaluation. We are excited to provide results for the requested experiments in the paper, and respond to individual comments below.
>
> **New experiments:**
>
> - Extension to distributional critics (Appendix I)
> - Extension to stochastic environments (Appendix J)
>
> **The authors should address methods in distributional RL (e.g. QR-DQN) and add it to the results.**
>
> DecQN is compatible with both ensemble Q function and distributional critics, and adding these features to DecQN would likely improve performance even further. We followed your suggestion and have added a new section on distributional RL to Appendix I that provides data for combining DecQN with a C51 head, as well as comparison to an IQN baseline. We chose IQN as it outperforms QR-DQN according to the original authors [Ref 1]. Our empirical results suggest that combining our decoupled agent with a distributional critic can further improve performance (Appendix I), particularly in challenging stochastic environments (Appendix J). DecQN + C51 significantly outperforms both DQN + C51 and the distributional IQN baseline. We note that non-distributional DecQN already matches performance of the distributional D4PG and DMPO actor-critic agents, while a more extensive investigation into decoupled distributional Q-learning provides promising avenues for future work.
>
> **The baselines are not up to date with other RL agents, such as SAC and DDPG.**
>
> We provide a brief comparison to SAC in Figure 10 of Appendix D and choose D4PG as a distributional extension of DDPG due to its stronger performance in [Ref 2]. Generally, we believe that several recent agents can achieve comparable performance, while our focus is to show that a very simple variation of the discrete action DQN agent can match state-of-the-art continuous action actor-critic performance on continuous control tasks.
>
> **Distributional Q-learning would be better to benchmark against than vanilla DQN.**
>
> The DMPO and D4PG baselines actually leverage distributional critics and are state-of-the-art on the corresponding tasks, while our DecQN agent directly builds on DQN with the major difference being a decoupled representation. Our core motivation was to keep the agent as simple as possible to highlight how minimal variations to DQN can yield surprisingly competitive performance compared to state-of-the-art actor critic algorithms.
>
> **Would there be similar observed benefits with different algorithms and replacing the Q-networks with the paper's proposed formulation?**
>
> We believe that decoupling in the value function would likely transfer to other value-based agents, while we focused on a basic Q-learning approach here to highlight how a very simple algorithm - without an explicit policy/actor - can be competitive with recent actor-critic algorithms.
>
> **Is there a failure case where DecDQN would fail when compared to other baselines?**
>
> The matrix games discussed in Section 5.1 provide some insights into where decoupling can yield coordination challenges but common continuous control tasks often alleviate at least some of these and still result in very competitive learning. To tease out potential failure modes, we have added new experiments for stochastic versions of Control Suite tasks to Appendix J, adding noise to either the observation or rewards. While we observe similar performance on most tasks, performance of DecQN is reduced on Humanoid Walk. Humanoid Walk combines several aspects that can hinder exploration, including implicit staged reward (first get up, then walk) and action penalties, which likely exacerbate coordination challenges when combined with stochasticity. We believe that special care should be taken when applying DecQN in environments that have the potential to amplify the coordination challenges observed in the simple matrix game domains of Section 5.1. Adding a distributional critic significantly improves performance in these stochastic settings and DecQN + C51 is able to match and even improve on the distributional D4PG agent. A more extensive investigation into decoupled distributional Q-learning provides promising avenues for future work.
>
> **Why does 21 bin Dec DQN generally do worse than 3 bin?**
>
> Increasing the number of bins increases the size of the action space and can pose additional exploration challenges with more options to try. Particularly in high-dimensional settings with implicit staged rewards and action penalties (Humanoid Walk), this can yield slower convergence rates. We have added this discussion to the main paper in Section 5.2.1. Furthermore, our new experiments on distributional RL in Appendix I (Figure 18) suggest that simply adding a C51 critic to DecQN without any parameter modifications/tuning significantly closes the gap for the 21 bin discretization on Humanoid Walk.

---

> > ### Author Response · Authors · 2022-11-15
> > **Response to Reviewer FMD4 (2 / 2)**
> >
> > **Is Sec 5.3 necessary? Dreamer-v2 would be slower due to being a model-based approach.**
> >
> > Section 5.3 primarily serves to show that agent performance extends to challenging visual control, yielding strong performance compared to recent state-of-the-art actor critic agents DrQ-v2 (model-free) and DreamverV2 (model-based), both with respect to sample and runtime efficiency. While DreamerV2 may be slower due to model-learning, it is still a very sample-efficient baseline.
> >
> > **Want to cite the library used to train other baselines (assuming it is RLlib)?**
> >
> > The training of DecQN, DQN, DMPO, and D4PG is based on the Acme library as specified in Appendix A - we have added a more prominent mention of this to the main paper in Section 5.2. Data for other baselines were either obtained directly from the original authors or by running their official code.
> >
> > Thank you very much again for your suggestions. We hope that our discussion and new experiments were able to sufficiently address any remaining concerns. Please do let us know during the discussion period if you have any questions and what would be required to reflect different levels of score improvement.
> >
> > **References**
> >
> > [Ref 1] W. Dabney, et al. "Implicit quantile networks for distributional reinforcement learning." ICML, 2018.
> >
> > [Ref 2] G. Barth-Maron, et al. “Distributed distributional deterministic policy gradients.” ICLR, 2018.

---

### Official Review · Reviewer_LrJN · 2022-10-24

**Confidence:** 4
**Correctness:** 3
**Technical Novelty And Significance:** 1
**Empirical Novelty And Significance:** 3
**Recommendation:** 8

**Clarity, Quality, Novelty And Reproducibility:**

The paper is clear and easy to follow
It's message is clear, and empirically verified
The paper presents limited novelty

**Strength And Weaknesses:**

Strengths:
- The paper is clear and easy to follow for the most part
- The authors perform a thorough empirical evaluation and empirically validate their claim that simple independent action discretization might perform as well as complicated actor-critic architectures
Weaknesses:
- Some terms used throughout the paper need to be introduced before being used, e.g. the authors continuously use the term bang-bang discretization without defining it
- The empirical evaluation is thorough but could be improved. I would have expected to see some domains where the 2 assumptions made by the authors are clearly invalid (independent action dimension and linearity of the value decomposition) and how this affects the performance of the DecDQN compared to the continuous control agents. I suspect that continuous control agents would behave also worse when action dimensions are not independent (since generally they apply diagonal covariance matrices like the authors underline) but we might have a different result when linear value decomposition is not valid, since this is an assumption heavily used by DecDQN. T

**Summary Of The Paper:**

The paper considers the problem of evaluating the performance of simple a simple variation of DQN (Deep Q- Networks), derived by discretizing actions across different dimensions independently, on standard continuous control tasks from the continuous control RL literature. By combining different ideas like value function decomposition and cooperative multi agent RL, the author are able to achieve performance comparable to algorithms designed for continuous control accross a variety of continuous control tasks.

**Summary Of The Review:**

Overall I am positive towards the paper. The claim made is simple and thoroughly investigated empirically. The paper does present limited novelty as the ideas discussed have been discussed before in the literature but In my opinion the thorough empirical evaluation adds to the papers contributions and ICLR is a perfect venue for this kind of results. Nevertheless, I would have also expected experiments in environments that explicitly break the 2 assumptions made to define DecDQN.

---

> ### Author Response · Authors · 2022-11-15
> **Response to Reviewer LrJN**
>
> Thank you very much for highlighting our strong empirical results, and for your suggestion of additional tasks. We are excited to present new experiments in the updated paper, and respond to individual questions below.
>
> **New experiments:**
>
> - Extension to distributional critics (Appendix I)
> - Extension to stochastic environments (Appendix J)
>
> **I would have expected to see some domains where the assumptions of independent action dimension and linearity of the value decomposition are clearly invalid and how this affects performance.**
>
> The assumptions of independent action dimensions and linearity of the value function actually do not hold throughout the continuous control benchmarks. Interestingly, the resulting approximation of the full problem characteristics are sufficient to generate highly competitive policies that yield performance at the level of state-of-the-art continuous control actor-critic algorithms. The discussion of coordination challenges in simple matrix games in Section 5.1 provides some intuition for how the decoupled agent is still able to efficiently solve the control tasks. However, the broad set of tasks solved by DecQN is still somewhat surprising.
>
> To better tease out potential failure cases we added **new experiments** with stochastic environments to Appendix J, where we add either observation or reward noise to the benchmarking tasks. While we observe similar performance on most tasks, performance of DecQN is reduced on Humanoid Walk. Humanoid Walk combines several aspects that can hinder exploration, including implicit staged reward (first get up, then walk) and action penalties, which likely exacerbate coordination challenges when combined with stochasticity. We believe that special care should be taken when applying regular DecQN in environments that have the potential to amplify the coordination challenges observed in the simple matrix game domains of Section 5.1.
>
> Adding a **distributional critic** significantly improves performance in these stochastic settings and DecQN + C51 is able to match and even improve on the distributional D4PG agent. A more extensive investigation into decoupled distributional Q-learning provides promising avenues for future work.
>
> **The authors continuously use the term bang-bang discretization without defining it.**
>
> We have added a short explanation of bang-(off-)bang control to the Decoupled Policies paragraph of the Related Works section including two references from the controls literature. We have also added a description of the resulting discretization process to Appendix A.
>
> Thank you again for your positive comments, and we hope that we were able to sufficiently address any remaining concerns. Please do let us know if you have any questions and what would be required to reflect different levels of score improvement.

---

> > ### Comment · Reviewer_LrJN · 2022-11-24
> > **Respond to authors**
> >
> > I thank the authors for addressing my concerns.
> > After some carefully checking the new empirical evaluations and some thought on the presented results, I have decided to raise my score to an accept. I think the paper provides enough empirical results to warrant a publication to ICLR

---

### Official Review · Reviewer_wU2o · 2022-10-27

**Confidence:** 4
**Correctness:** 3
**Technical Novelty And Significance:** 2
**Empirical Novelty And Significance:** 3
**Recommendation:** 6

**Clarity, Quality, Novelty And Reproducibility:**



The method of the paper is very clear. The experiments are solid and sufficient to verify the effectiveness of the method. The paper conduct experiments on various benchmark tasks, including the visual input task.

Applying the decentralized method in MARL is very intuitive. I'm surprised that this method is never proposed before.

The implementation of the paper is not open-sourced. Open-source code is not compulsory, however, it could help readers understand some details of the methods.





**Details Of Ethics Concerns:**



**Strength And Weaknesses:**



Strength:

The method of the paper is very clear. The method of applying the decentralized technique to the continuous action space problem is very intuitive. The experiment results are also sufficient.



Weakness:

The proposed method is quite simple. The author only considered the average aggregation function for the decoupled Q values. As the aggregation method is widely discussed in MARL literature and there still remains some performance gap with the state-of-the-art actor-critic methods (e.g. Humanoid Stand) , I think the paper should consider using a more complex aggregation function.

Minor typos:

- $|\mathcal{A}|$ should be $\mathop{dim}(\mathcal{A})$.





Several Questions:

- I don't get the necessity of comparing Dreamer in Section 5.3.

- Do you consider using distribution to model the discretized action, as what is done in bang-bang policy?

- In MARL, some works (e.g. QMIX) used a monotonic function to aggregate the Q values for each decentralized action. However, it's interesting to see that the author only uses the average function (eq. 4) and performs well. Do you ever try some other monotonic aggregation function?
- Can you provide the range of the values for each dimension on some of the tasks? Dies the range vary from different dimensions?

**Summary Of The Paper:**





This paper extended Q-Learning to continuous action space by the action discretization with value decomposition in MARL. The experiment results show that the proposed method is competitive with the state-of-art continuous actor-critic methods.



**Summary Of The Review:**



The proposed method in this paper is very simple and effective. Applying the decentralized method in MARL is very intuitive. I'm surprised that this method is never proposed before.

However, as the aggregation method is widely discussed in MARL literature and there still remains some performance gap with the state-of-the-art actor-critic methods (e.g. Humanoid Stand), I think the paper should consider using a more complex aggregation function in eq. 7.

---

> ### Author Response · Authors · 2022-11-15
> **Response to Reviewer wU2o**
>
> Thank you very much for appreciating the simplicity and effectiveness of our method, as well as for your feedback and suggestions. We are excited to present new experiments in the paper, and elaborate on each of the questions below.
>
> **New experiments:**
> - Extension to distributional critics (Appendix I)
> - Extension to stochastic environments (Appendix J)
>
> **Why is it necessary to compare to Dreamer in Section 5.3?**
>
> For visual control experiments, we chose DrQ-v2 and DreamerV2 as the current state-of-the-art model-free and model-based agents with the strongest performance on these benchmarks. DrQ-v2 is generally more runtime-efficient than DreamerV2 while being less sample-efficient on some tasks. Furthermore, DrQ-v2 uses some task-specific variations of the hyperparameters, while DreamerV2 and DecQN each use a fixed set of hyperparameters. Our goal is to show that DecQN yields strong performance compared to both model-free and model-based agents, where DreamerV2 is a very strong as well as sample-efficient baseline that also uses a fixed set of hyperparameters.
>
> **Do you consider using distribution to model the discretized action, as what is done in bang-bang policy?**
>
> Distributional critics are compatible with our approach and we have added new experiments and a discussion to Appendix I and J. We combine both DecQN  and DQN with a C51 head and compare to IQN. The distributional critic slightly improves performance at convergence while sample-efficiency of DecQN + C51 is on average similar to DecQN (Appendix I). The distributional critic can significantly improve performance and help overcome coordination challenges in stochastic environments (Appendix J). A more extensive investigation into decoupled distributional Q-learning provides promising avenues for future work, while our primary motivation here was to show that state-of-the-art continuous control performance can be obtained with a surprisingly simple variation of regular DQN.
>
> **Do you ever try some other monotonic aggregation function?**
>
> We did try more complex aggregation functions based on QMIX, and briefly comment on this at the end of Section 4.2. Similar to Tavakoli et al. (2021), we did not observe substantial improvements and introducing more complex, e.g. learned, aggregation functions comes at the risk of additional tuning or sample complexity. Our core motivation was to keep the agent as simple as possible to highlight how minimal variations to DQN can yield competitive performance compared to state-of-the-art actor critic algorithms. Furthermore, our new experiments on distributional RL in Appendix I (Figure 18) suggest that simply adding a C51 critic without any parameter modifications/tuning closes the remaining gap on tasks where regular DecQN did not perform at least as good as the best baseline (e.g. Humanoid Stand), further underlining the strength and versatility of this very simple approach
>
> **Can you provide the range of the values for each dimension on some of the tasks? Does the range vary from different dimensions?**
>
> Both the DeepMind Control Suite and MetaWorld use action spaces with boundaries -1 <= a <= +1 (Quadruped uses -0.8<= a <=+0.8 for some dimensions). For MiniCheetah, we use -a_bound <= a <= +a_bound around the default joint positions. For simplicity, we then simply use evenly spaced bins between the extreme values, while any mapping from indices to action values is supported. We have added a description of the discretization process to Appendix A.
>
> **The implementation of the paper is not open-sourced, it could help readers understand some details of the methods.**
>
> We have followed your suggestion and uploaded anonymized code as a .zip file under [this link](https://drive.google.com/file/d/1imAGdwDJKdWOnKP7hE46vnvqNLb6Pz_s/view?usp=sharing). We will further link directly to a GitHub repository for ease of access upon publication.
>
> **Minor typo**
>
> Thank you for catching this, we have updated |space| to dim(space) throughout.
>
> Thank you again for your comments and interest, and we hope that our discussion and new experiments were able to sufficiently address any remaining concerns. Please do let us know if you have any questions and what would be required to reflect different levels of score improvement.

---

### Author Response · Authors · 2022-11-17
**Additional experiments and revisions**

We would like to thank the reviewers again for their positive feedback and suggestions for improvement.

We have tried to address all comments individually, with corresponding revisions and additions to the paper in blue. The updates include a new line of experiments on distributional RL and stochastic environments in Appendices I & J.

If there are any remaining questions, please do not hesitate to ask. Thank you very much!

---

### Decision · Program_Chairs · 2023-01-20

**Decision:**

Accept: poster

**Justification For Why Not Higher Score:**

The novelty is below the bar of spotlight.

**Justification For Why Not Lower Score:**

The results are impressive.

**Metareview: Summary, Strengths And Weaknesses:**

This paper presents a critic-only approach to solving continuous control problems, by combining value decomposition and action discretization. They test this approach across a large set of environments from continuous control benchmarks and show that this simple model-free approach performs on par with SOTA actor-critic methods, both for model-free and model-based methods, and based on state features and pixel observations.
Reviewers raised concerns about the experiments. The concerns were addressed during the rebuttal period. Overall this is a solid paper, and the AC recommends acceptance.

**Note From Pc:**

if the above contains the word "oral" or "spotlight" please see: "oral" presentation means -> notable-top-5% and "spotlight" means -> notable-top-25%. As stated in our emails, we are disassociating presentation type from AC recommendations